# The Impact of Different Types of Social Resources on Coping Self-Efficacy and Distress During Australia’s Black Summer Bushfires

**DOI:** 10.3390/ijerph22091341

**Published:** 2025-08-27

**Authors:** Greta Amorsen, Jacki Schirmer, Mel R Mylek, Theo Niyonsenga, Douglas Paton, Petra Buergelt, Kimberly Brown

**Affiliations:** 1Health Research Institute, Faculty of Health, University of Canberra, 11 Kirinari Street Bruce, Canberra, ACT 2617, Australia; jacki.schirmer@canberra.edu.au (J.S.); mel.mylek@canberra.edu.au (M.R.M.); theo.niyonsenga@canberra.edu.au (T.N.); kimberly.brown@canberra.edu.au (K.B.); 2Centre for Environmental Governance, Faculty of Business, Government and Law, University of Canberra, 11 Kirinari Street Bruce, Canberra, ACT 2617, Australia; 3Faculty of Health, University of Canberra, 11 Kirinari Street Bruce, Canberra, ACT 2617, Australiapetra.buergelt@canberra.edu.au (P.B.)

**Keywords:** social capital, social resources, disaster response, bushfire, wildfire, psychosocial stress, psychological distress, self-efficacy, individual resilience

## Abstract

While social resources are known to promote positive psychological outcomes after disasters, little is known about the unique influence of different social resources on distress and coping during a disaster. This study examined the association between five social resources: sense of belonging, bushfire reciprocal support, emotional support, practical support and loneliness, and two psychological outcomes, distress and coping self-efficacy, during Australia’s 2019–2020 Black Summer bushfires. Survey data collected from 2611 bushfire-affected Australians in late 2020 was analysed using regression modelling. Higher perceived emotional and practical support and lower levels of loneliness predicted increased coping self-efficacy, and higher sense of belonging and lower loneliness predicted reduced distress. However, higher emotional and reciprocal support predicted higher distress after accounting for coping self-efficacy. The findings suggest having higher access to some social resources may not directly reduce distress but may reduce distress indirectly through increasing coping self-efficacy. While access to social resources, particularly bonding social capital, is likely important for supporting psychological response during disasters, the findings suggest this may be dependent on the perceived quantity, quality and expectations of these social resources. The findings indicate that different social resources interact with disaster-related psychological outcomes in distinct, complex and sometimes non-linear ways.

## 1. Introduction

Natural hazard events such as bushfires are increasing in frequency and severity worldwide as a result of climate change [1,2]. This increases the risk of natural hazards becoming disasters, defined as natural hazard events that have significant negative impacts on human populations [3]. It is well recognised that both investing in disaster preparedness and providing supports post-disaster reduces these negative impacts. In addition, supporting at-risk populations to respond effectively during and immediately after a disaster can save lives and reduce the severity of physical and psychological impacts [4,5,6]. In this paper, we examine whether having access to five types of social resources supports positive psychological outcomes during and immediately after disaster, with a focus on two psychological outcomes known to be particularly critical to proactive and effective action: coping self-efficacy and psychological distress.

There is strong evidence that experiencing high levels of distress during and immediately after a disaster can negatively affect both disaster response and recovery. ‘Immediately after’ refers to what is sometimes called the ‘early recovery’ stage, lasting anywhere from one to several months and in which fires are no longer active, but residents are grappling with the first stages of coping with the damage experienced [7]. Exposure to disasters is commonly associated with higher rates of psychological distress, including a loss of sense of control and safety, increased levels of stress and anxiety and sleep disruptions [8,9,10,11,12]. Experiencing some stress can be important for motivating protective action. However, experiencing heightened and prolonged stress levels—defined here broadly as ‘distress’—can cause feelings of helplessness and impair decision making and cognitive abilities, thus impacting short and long-term mental health and coping abilities during and after a disaster [13,14,15]. Inversely, high psychological wellbeing has been associated with improved cognitive functioning [16]. However, the causality of these associations is likely complex (see for example [17,18,19]).

Multiple theories suggest that having access to resources is crucial for supporting functioning in response to stressors such as disasters. For example, Hobfoll’s Conservation of Resources (COR) theory suggests that distress is caused by a perceived loss of resources which arises from an imbalance between the demands experienced by a person and the resources they have available to assist coping [20,21,22,23]. Having access to high levels of relevant resources can protect against negative impacts of stress and result in better functioning and psychological outcomes [12,20,24].

The conceptualisation of stress as a function of demand relative to resources aligns with Social Cognitive Theory (SCT), which suggests that people’s stress reactions are influenced by their coping appraisal, defined as their perceived ability to cope with stressors in their environment [25]. A key psychological resource that influences people’s coping appraisal is self-efficacy, defined by Bandura as an individual’s belief that they can exert control over or take action in relation to events in their life [25,26,27]. Self-efficacy has been found to influence motivation, cognition and decision processes in relation to a range of behaviours and stressors [28,29,30,31]. Those with higher levels of self-efficacy are more likely to take proactive coping approaches, as well as to feel a greater sense of control and a greater sense that they are able to manage the negative impacts of stressors [29].

The theories of COR and SCT can support an understanding of how levels of distress are likely to predict a person’s perceived ability to cope during and after a disaster. In this paper, we refer to this perceived coping ability as a specific form of self-efficacy referred to here as ‘coping self-efficacy’. Having high levels of self-efficacy in one’s ability to cope with stressful events has been associated with reduced distress and an increased sense of control and proactive coping, which are important for supporting adaptive behavioural responses before, during and after a disaster [29]. Studies examining the role of self-efficacy in disasters have found self-efficacy is associated higher levels of disaster preparedness knowledge and behaviour, and with evacuation intentions [32,33,34]. More specifically, response efficacy, the belief that one’s behaviour can reduce threat levels, can facilitate adaptive behaviours such as leaving early in response to a bushfire [35,36]. Conversely, low levels of response efficacy and self-efficacy can lead to maladaptive behaviours such as delayed decision making when faced with a threat [35]. Self-efficacy has also been associated with reduced psychological distress after disasters and increased perseverance in recovering from losses post-disaster [29,37,38,39]. Overall, the evidence suggests that having high coping self-efficacy is likely to contribute to reduced distress—while also being dependent on the level of distress a person experiences as a result of disaster.

Having lower distress and higher coping self-efficacy can increase the likelihood that a person will be able to make the right decisions to protect themselves and their loved ones during a disaster, and cope with the potential impacts in the days and months after it occurs. To support these positive outcomes, it is important to identify the factors that are most likely to contribute to both lower distress and higher coping self-efficacy during and immediately after disasters. In general, it is well accepted that people’s ability to effectively respond to, cope with and recover from stressors such as disasters depends on their access to a wide range of resilience resources [20,40,41,42,43,44]. These include financial resources (e.g., insurance, savings), institutional resources (e.g., good governance, functioning government support systems), human and psychological resources (e.g., education, skills, physical and mental health), and social resources (a person’s social networks and access to social support).

In this study, we examine the role of social resources, which are widely documented in the literature as supporting positive disaster outcomes [9,45,46,47]. Both SCT and COR theory suggest that social resources are critical to protecting against resource loss and promoting psychological resources that assist in coping with disasters [20]. For example, having higher perceived social resources can lower the perceived potential for harm caused by a stressor such as a disaster [48]. Social resources can also reduce stress by supporting self-efficacy and encouraging active coping strategies such as problem solving [49,50]. Receiving social support can also increase people’s coping appraisal, thus increasing their self-efficacy and reducing stress and stress symptoms such as negative affect [48,51].

Social resources, also known as social capital, include a person’s number and types of social networks, the resources or support, and sense of belonging or connection those networks provide [52]. Having good access to social resources has been found to support disaster preparedness [43] and positive decision making related to evacuation [53,54,55,56], as well as to buffer against distress and other negative mental health outcomes after disasters [41,46,47].

There are gaps in knowledge about the role of social resources in contributing to positive outcomes during the disaster response phase. Many studies examine changes in access to social resources after disasters [41], or the contribution of social resources to positive behavioural outcomes before and during disasters [53] and improved long-term disaster recovery [46]. However, few have examined the role of social resources during the disaster response stage other than in evacuation behaviours, or in the early recovery stage. Additionally, less is known about whether specific types of social resources are more important for supporting coping self-efficacy or reduced distress during a disaster. Previous studies have examined a wide range of types of social resources, from a person’s social networks [46] to a person’s access to social support [47], as well as the psychological sense of connection to others [57]. However, very few studies have explicitly examined the differential roles of these types of social resources in contributing to psychological outcomes during disaster response and early recovery.

This paper contributes to addressing these gaps by investigating whether access to five different types of social resource predicts variance in coping self-efficacy and psychological distress levels during and immediately after disasters. We analyse a sample of 2611 individuals directly impacted by Australia’s 2019–2020 Black Summer bushfires who completed the 2020 Regional Wellbeing Survey (RWS). We examine whether access to five types of social resources predicted higher coping self-efficacy during the bushfires and/or lower distress after controlling for the effects of coping self-efficacy.

We first briefly review types of social resources and current evidence for their role in disaster resilience. We then describe the study methods and present descriptive findings followed by regression modelling. Our discussion considers the implications of the findings for understanding both the role of social resources overall in supporting positive psychological outcomes, and the role of different types of social resources.


*Social Resources and Disaster Resilience*


A large body of research has examined the role of social resources in disaster resilience [52,58,59]. Common findings include that better access to social resources is associated with the following:Better mental health outcomes after disasters [41,46,47,57,60,61];Increased rates of recovery after disasters [58,59];Higher levels of engagement in preparedness actions before disasters [34,43,53];Increased likelihood of evacuating to a safe place during disasters [54,56,62].
*Defining types of social resources: cognitive vs. structural, received vs. perceived*

A diversity of types of social resources have been examined in past disaster studies, including the number and types of social connections a person has (their social networks) [46,62,63], a person’s perceived and received access to support via their social networks [57,64], or less commonly, levels of social cohesion [57]. Social resources are often differentiated into structural and cognitive resource types [65]. Structural social resources refer to the types of social networks, or the number of connections and interactions people have, including reciprocal support exchanges. Social resources are commonly differentiated based on whether they involve bonding, bridging, or linking capital. Bonding capital refers to close social connections, usually between friends and family; bridging capital refers to connections between social groups such as members of an organisation or community; and linking capital refers to connections between individuals and civic agencies and those in power [52]. Social networks can provide access to a range of resources, sometimes referred to as ‘network social capital’ [66], including information, practical and emotional support. They can also create a sense of social support, belonging and trust, and reduce feelings of loneliness. These are examples of cognitive social resources, a person’s beliefs, perceptions and attitudes about their social environment [41,67,68].

The presence of both structural and cognitive attributes leads to complexity in measuring social resources. For example, while loneliness or sense of belonging are considered cognitive social resources based on a person’s perceived sense of connection to others, social isolation is often considered a structural resource, measured based on assessing the structure and density of people’s social networks and frequency of social interaction [66]. While one is structural and the others are cognitive, social isolation and sense of loneliness and belonging are strongly associated with each other.

There is an ongoing debate about the extent to which a person’s different social resources have distinct roles, versus them being proxies for a single underlying ‘social resource’ a person has access to. There is growing evidence that different social resources are ‘related but distinct’, with sufficient differentiation in their prevalence and association with outcomes such as health to suggest the presence of a range of unique social resources with differentiated roles [69]. For example, Moore and Carpiano [66] found mostly weak associations between five types of social resources (generalised trust, trust in neighbours, network diversity, social isolation and social participation), the differential presence of these resources in women and men and differential associations with health outcomes. Previous studies have also found that perceived access to social support is often more strongly related to psychological outcomes compared to receiving support during stressful events, and found low correlations between received and perceived social support [41,64,70]. This suggests a need to understand whether different types of social resources vary in their strength and type of association with disaster-related psychological outcomes.

A comprehensive review of the varied types of social resources examined in past disaster studies is beyond the scope of this paper. However, some common findings are briefly reviewed below. Previous disaster studies have examined the types of resources people access through their social networks, and the disaster-related norms and attitudes they are exposed to via their social networks (see for example [46]). Some studies have identified that cognitive social resources, including social cohesion, sense of community, place attachment and self- and collective efficacy, are positively associated with disaster preparedness actions [34,43,53]. Social networks that support sharing of knowledge, resources and strategies have been found to facilitate better physical and psychological disaster preparedness, leading to reduced negative impacts [71].

Many studies have found that social networks serve as a crucial resource for information and warnings during a disaster. People often first hear about disaster warnings through friends, family and neighbours and seek further information and guidance from them to inform their response [4,72]. These social networks can encourage positive response behaviours such as making informed decisions to evacuate early, or to stay and defend a well-prepared property [44,54,62,73,74,75]. The ways in which different types and sources of crisis communication impact individual disaster response behaviours have been examined in a growing number of studies, although they are not the focus of the current study [72,76].

Social networks can also be sources of immediate aid and practical support, such as transport and accommodation, to assist in timely evacuation and securing people’s safety [52,58,77,78]. A smaller number of studies have identified that having access to and receiving social support can assist people in physically responding after a disaster in ways other than evacuation [41,57,79]. Overall, social networks appear to be important sources of information and practical support during disasters and may be associated with more positive psychological outcomes such as reduced risk of distress and higher coping self-efficacy. However, past studies have not directly assessed linkages between access to practical support and these specific psychological outcomes.

The findings of past studies also suggest that social networks must have positive characteristics to be effective in promoting more positive outcomes before, during and after disasters. For example, having a supportive and trustworthy social network has been found to promote coping self-efficacy in disaster recovery [80,81]. Perceived access to support from one’s social networks is important. As supported by the theories of SCT and COR, feeling confident in access to social support is associated with higher coping self-efficacy and reduced distress [20,21,25]. Some studies have found that higher levels of perceived social support were associated with reduced psychological distress in short-term recovery [47,73], and with reduced risk of post-traumatic stress and depressive symptoms among disaster-exposed individuals [82,83]. However, these expected associations have not always been found, with some studies finding no association between perceived access to social support and stress or distress levels [84,85].

Social networks enable both receipt and provision of support, with reciprocal exchanges being key features of most. However, reciprocal support exchange has been less studied in relation to disaster psychological outcomes than received or perceived availability of support from networks [79]. Previous studies suggest that the level and type of support exchanged depends on the level of disaster impact experienced. Those more greatly impacted are often more likely to receive rather than provide support, and to have higher expectations of receiving support—as well as to have higher levels of distress, likely at least partly due to the level of impact experienced [80,81,86]. However, receiving support may also increase perceived support availability, leading to better coping. Some studies identify that reciprocal support provision during and after disasters can increase disaster-related self-efficacy [80,81,87].

Overall, existing studies point to the potential for different social resources to have distinct associations with psychological outcomes such as distress and coping self-efficacy. However, research is needed to identify how different forms of social resource influence psychological outcomes, as most studies tend to examine a single social resource rather than examining the unique roles of different social resources, including both structural and cognitive social resources. Most previous studies have examined associations between social resources and disaster preparedness or longer-term recovery, with a gap in understanding of the association between access to different types of social resources and coping and distress during and immediately after disasters. This study begins to address this gap by examining the association between five social resources, including practical and emotional support, bushfire reciprocal support, sense of belonging and loneliness. Based on previous findings, we hypothesised that higher levels of access to these resources, and lower levels of loneliness, would be associated with higher levels of coping self-efficacy and lower levels of distress during and immediately after the 2019–2020 bushfires.

## 2. Materials and Methods

We examined the role of social resources in predicting self-reported distress and coping self-efficacy during the disaster response and immediate recovery phase through analysing data collected in the 2020 Regional Wellbeing Survey (RWS). The RWS began in 2013 (Wave 1) and is an annual, nationwide survey asking adult Australian residents about multiple topics, including preparation for and response to disasters. Each wave has between 10,000 and 20,000 respondents. It is primarily a cross-sectional survey, with a sub-section of the sample tracked longitudinally over time. The 2020 RWS (Wave 8) included specific questions on experiences of Australia’s 2019–2020 Black Summer bushfires, as well as a number of questions examining access to social resources.

### 2.1. Context and Study Area

The 2019–2020 Black Summer bushfires occurred during Australia’s hottest and driest years on record, starting in mid-2019 and continuing until March 2020. New South Wales (NSW), Victoria (VIC), the Australian Capital Territory (ACT), Queensland (QLD) and South Australia (SA) had the largest geographic areas affected by the bushfires, and at times, multiple bushfires were burning across all eight Australian states and territories simultaneously [1,88]. The bushfires directly impacted many communities, burning over 24 million hectares, destroying 3100 houses and directly causing 33 deaths [1,89]. The extent of the bushfires meant that the RWS sample included people impacted by the fires who lived in many different locations across Australia.

### 2.2. Survey Distribution

The 2020 RWS was conducted between October 2020 and January 2021 (approximately one year after the start of the 2019–2020 Black Summer bushfires) and was open to people aged 18 years and over residing in Australia at the time of survey completion. The survey could be completed online or using a paper form, and multiple methods of recruitment were used. Past RWS participants were invited to complete the survey again if they had given permission to be contacted in future. In addition to this, survey invitations were posted to randomly selected addresses across Australia, with addresses obtained from the Geoscape National Address File (G-NAF), which includes every address in Australia. Regions impacted by the 2019–2020 Black Summer bushfires in NSW and Victoria were more intensively sampled to increase the sample of those affected by the bushfires. Impact was defined based on whether the Australian Tax Office (ATO) declared the local government area (LGA) bushfire-affected in the 2019–20 financial year (see Figure 1 for declared bushfire-affected LGAs), a declaration used to provide tax support to people in areas where there was a likelihood of having experienced impacts from the bushfires [90]. Sampling from the G-NAF was stratified to oversample areas that were declared bushfire-affected during the Black Summer bushfires. The RWS also deliberately over-samples rural and remote regions of Australia compared to urban regions, and farmers relative to other occupation groups.

Online panel recruitment and social media advertisement were used to increase responses from younger people, and from those living in urban areas. This was undertaken to address known bias in usual responses from mailing recruitment, which is typically biased towards older respondents, and social media was used to promote the survey more broadly. Overall, this combination of sampling methods sought to achieve a large sample across Australia, including a large sample from bushfire-impacted regions. In total, 21,939 people participated in the 2020 RWS.

As the RWS uses multiple reinforcing recruitment methods, it is not appropriate to calculate an overall response rate; additionally, there is widespread recognition that response rates are poor indicators of survey sample quality [91]. Instead, characteristics of the sample as a whole were compared to data from the 2021 Australian Census of Population and Housing [92]. In addition to deliberate over-sampling of bushfire-impacted regions, rural/remote areas and farmers, the sample was biased towards women and older people, which is common in many surveys [93]. Both deliberate sampling bias (bushfire-impacted regions, rural/regional, farmers) and unintended sample bias (gender, age) were addressed in analyses through actively controlling for them in modelling, as described further in the Results section. For this study, a subset of the national sample was analysed, as described below.

### 2.3. Sample

The sample analysed in this study consisted of people who either (i) self-identified as having been affected by the 2019–2020 Black Summer bushfires in response to the question, ‘Overall, how much were you or your household affected by bushfires in the last 12 months?’ or (ii) indicated they experienced one or more direct impacts from the bushfires (for example, their house was damaged, destroyed or at risk of damage, their car or other property was damaged or destroyed, they were injured or had an illness due to bushfire/smoke, they evacuated, or they defended their home).

Of the 21,939 who completed the 2020 RWS, 2611 met the criteria for inclusion in analysis based on (i) reporting experiencing bushfire impacts, (ii) being asked all survey items analysed in this paper, or (iii) not answering ‘NA’ or ‘don’t know’ to any variables analysed in this paper (those who answered NA or don’t know were excluded as only those who were able to rate their experiences and felt that questions applied to them were considered valid respondents). Analysis using Little’s MCAR test revealed data was not missing completely at random (X^2^ = 344.370, *p* = 0.00). Listwise deletion of missing data was used because missing data on all variables was less than 6%, a large sample size remained (n = 2611) and the slightly non-normal distribution of outcome variables may have increased bias if missing data imputation was used [94].

The final valid sample was aged 18 to 94 (M = 57.47, SD = 14.44), and included 1560 females (59.7%) and 1051 males (40.3%). The distribution of the sample across different states and territories reflected the distribution of the Australian population and bushfire impact, with the majority residing in NSW (47.2%) and VIC (31.3%), while fewer resided in QLD (7.7%), SA (6.0%), TAS (3.1%), WA (2.7%), ACT (1.1%) and NT (0.8%). Figure 1 displays the distribution of the survey sample by LGA and location of ATO bushfire-affected LGAs [90]. Table 1 shows the demographic characteristics of the sample.

### 2.4. Measures

Table 2 summarises the measures examined in this paper, while Table 3 presents the results of Exploratory Factor Analysis (EFA) for three measures where no existing validated scales existed. This includes the two outcome variables examining the psychological outcomes of distress and coping self-efficacy experienced during the time the bushfires were active. These three measures are also described in Table 2 and refer to the EFA where relevant. Five social resources were examined, using data available from the survey:Emotional support—a retrospective measure of perceived access to emotional support during the time the bushfires were active, measured using a single item.Practical support—a retrospective measure of perceived access to practical support during the time the bushfires were active, measured using a single item.Sense of belonging (existing scale)—a cognitive social resource that asks about sense of belonging at the time the survey was completed.Loneliness index—a cognitive social resource often associated with strength and depth of social networks, measured using existing validated scale and asking about loneliness at the time the survey was completed.Bushfire reciprocal support—a retrospective measure of reciprocal help given to and received from neighbours and other community members during the fires; as this was a new measure, EFA was used to construct a scale.

It is important to note that the survey did not measure all types of social resources: instead, we analysed five social resources asked about in the survey that had potential relevance to understanding distress and coping self-efficacy. It is likely other types of social resources may have influenced how people responded during the bushfires; however, our analysis was constrained by the data available in the RWS. In addition to social resources, potential confounding variables were included in the analysis, including self-rated level of disaster impact, age and gender (Table 2).

EFA was used to determine whether individual items related to experiences during the bushfires that were thought to be measuring underlying constructs, and could be combined into scales measuring (i) distress, (ii) coping self-efficacy and (iii) bushfire reciprocal support. All items listed in Table 3 were included in one EFA. An initial examination of the correlation matrix found no high correlations above 0.60, indicating no likely problems associated with multicollinearity. A principal axis factoring approach with an oblique rotation (oblimin) was used [99]. The Kaiser–Meyer–Oklin value was KMO = 0.84 [100], and *p*-value for Bartlett’s test of sphericity was <0.001, suggesting all items were factorable [101]. The items entered formed three factors, as shown in Table 3.

### 2.5. Data Analysis

Data analysis was conducted using IBM Statistical Package for Social Sciences (SPSS) version 29. Data analysis included (i) exploratory factor analysis (EFA), (ii) descriptive statistics summarising and exploring the data at hand, including the correlations between scale items used for EFA, and (iii) regression modelling.

Two hierarchical multiple regression analyses were conducted to examine the contribution of social resource variables to coping self-efficacy and distress during the bushfires. The analysis controlled for demographic factors, including age and gender, which have been found in previous studies to predict variation in disaster response and distress associated with this response. Studies report that women experience higher levels of distress after disasters [73] and have greater perceived susceptibility to threat [102], while men are often more confident in their coping abilities [103]. Similarly, older people are more likely to have prior experience with a natural hazard, which is associated with increased preparedness, natural hazard knowledge and likelihood of evacuating [44,61,104]. Participants’ self-rated levels of impact from the bushfires were also controlled for, as disaster exposure is often cited as a strong predictor of psychological outcomes after disasters and is likely to impact access to social resources [6,9,41,49].

We acknowledge that there is some debate around the use of single Likert scale items in linear regression. However, to determine the unique influence of emotional and practical support, we deemed it necessary to include them as single items. Based on the literature, we believe it was appropriate to include these single-item predictor variables [105,106].

Data screening revealed the outcome variables were slightly non-normally distributed, with coping self-efficacy negatively skewed (−0.50) and distress only slightly negatively skewed (−0.16). However, with the large sample size, the regression models were judged to be robust against non-normality [107,108]. The sense of belonging scale had several univariate outliers and was slightly negatively skewed (−1.08). However, an inspection of histograms and QQ plots showed most variables were approximately normally distributed.

The first regression model predicted coping self-efficacy (average of three items measured on 7-point scale) and model 2 predicted distress during bushfire (average of six items measured on 7-point scale). The variables controlled for in step 1 of each model included age, gender (dummy coding: female = 0, male = 1) and self-rated level of bushfire impact. The social resource variables added in step 2 of the models included practical support and emotional support (single-item measured on 7-point scale), bushfire reciprocal support (average of three items measured on 7-point scale), sense of belonging (average of three items measured on 7-point scale) and loneliness (average of three items measures on a 5-point scale). Coping self-efficacy was also included as a predictor variable in model 2.

## 3. Results

Table 4 presents descriptive statistics for dependent and independent variables. Response categories of low, moderate and high are only used for reporting descriptive statistics and not used in the subsequent regression analyses. Over half of the participants (54.0%) reported high levels of coping self-efficacy during the bushfires and almost half (40.4%) reported low levels of distress. More participants reported having access to high levels of emotional support (61.6%) during the bushfires compared to practical support (47.7%) or bushfire reciprocal support (44.4%). The majority of the participants also reported high levels of sense of belonging (76.8%) and low levels of loneliness (63.8%). More than half of the participants (49.4%) reported high levels of self-rated bushfire impact.

Correlations between the five social resources and the two outcome variables were explored to understand whether the five resources had similar associations with each outcome, as well as correlations between each type of social resource and demographic variables controlled for in regression analysis (Table 5). Spearman’s correlation was used as the analysis contained ordinal data. This showed differing and variable associations. Higher levels of social resources (higher practical support, emotional support, bushfire reciprocal support and sense of belonging, and lower loneliness) were associated with higher coping self-efficacy, albeit with less strong associations for bushfire reciprocal support and sense of belonging. Associations with distress varied as distress was high amongst those with higher loneliness, lower sense of be-longing and higher levels of bushfire reciprocal support. Access to practical and emotional support was not strongly associated with differing distress levels. Associations between the five social resources were significant but were generally only weakly to moderately correlated (all but two having correlation coefficients more than 0.3, but most less than 0.5), consistent with the hypothesis that the resources are functionally different from each other.

### 3.1. Coping Self-Efficacy and Distress During Bushfires

We examined whether the level of perceived and received access a person had to different types of social resources was associated with a difference in coping self-efficacy or distress during bushfires, using two regression models. The first model examined whether access to the five social resources predicted variation in coping self-efficacy. The second then examined whether access to these five social resources predicted distress during bushfires, while controlling for the effect of coping self-efficacy. The two models were examined in this way as available evidence suggests that access to social resources may predict differences in higher coping self-efficacy and/or distress, but it is unclear whether some predict one but not both of these. Additionally, as self-efficacy is a known predictor of lower distress, there was a need to control for its effects when examining the association between access to social resources and distress. Both models included age, gender and self-rated bushfire impact as confounders in the first step of the model, given the known associations between these and the outcome variables.

The tolerance (>0.10) and VIF (<10) statistics were within recommended limits [109] for both regressions and examination of the correlation coefficients suggested there was no multicollinearity. The Durbin–Watson statistic for the distress model (2.01) and the coping self-efficacy model (2.03) indicated independence of residuals and Cook’s D values indicated there were no influential cases (<0.1). A total of 10 multivariate outliers were detected through an examination of Mahalabonis’ distance values in the distress model and two were detected in the coping-self-efficacy model. However, these outliers were not removed as they did not significantly change the results. Histogram and normal probability plots of the standardised residuals and scatterplots of the standardised residuals against the predicted values suggested that normality, linearity and homoscedasticity assumptions were met for both regression models [107].

### 3.2. Model 1: Coping Self-Efficacy During the Bushfires

The first regression model examined variation in coping self-efficacy during the bushfires (Table 6). At step one of the model, age, gender and self-rated bushfire impact accounted for 7.3% of the variance in coping self-efficacy during the bushfires (Δ*F*(3, 2607) = 68.25, *p* < 0.001). The addition of practical support, emotional support, bushfire reciprocal support, sense of belonging and loneliness in step two of the model explained an additional 16.8% of the variance (Δ*F*(5, 2602) = 114.83, *p* < 0.001). The final model predicted 24.0% of the variance in coping self-efficacy (*F*(8, 2602) = 102.99, *p* < 0.001), with adjusted R^2^ = 0.24 and a large effect size (*f* = 0.56) [110].

In the final step of the regression model, gender was the largest significant predictor of coping self-efficacy (with males reporting higher coping self-efficacy compared to females), followed by emotional support, practical support, self-rated bushfire impact and loneliness. The presence of social networks that could be relied upon for emotional and practical support was positively associated with coping self-efficacy during the bushfires, and higher levels of loneliness were associated with lower coping self-efficacy during the bushfires. Sense of belonging and bushfire reciprocal support were not significant predictors in the final model.

### 3.3. Model 2: Distress During the Bushfire

The second model examined associations between social resources and distress during the bushfires, while controlling for the known positive effect of coping self-efficacy on distress (identified in previous studies) (Table 7). At step one, age, gender and level self-rated bushfire impact accounted for 20.3% of the variance in distress during the bushfires (Δ*F*(3, 2607) = 221.92, *p* < 0.001). The addition of the five social resource variables (practical support, emotional support, bushfire reciprocal support, sense of belonging and loneliness) in the final model accounted for a further 13.7% of the variance (Δ*F*(9, 2601) = 89.94, *p* < 0.001). The final model predicted 34.0% of the variance in distress during the bushfires (*F*(9, 2601) = 149.08, *p* < 0.001), with adjusted R^2^ = 0.34 and a large effect size (*f* = 0.72) [110].

In the final step of the model, coping self-efficacy was the strongest predictor of distress during the bushfires, followed by self-rated bushfire impact, loneliness, bushfire reciprocal support, gender, emotional support and sense of belonging. All the predictors in the final step were statistically significant except for practical support. Sense of belonging had a negative relationship with distress during bushfires. However, it accounted for a relatively small proportion of its variance. Emotional support, bushfire reciprocal support and loneliness were positively associated with distress during bushfires, and as expected, coping self-efficacy had a negative relationship with distress.

## 4. Discussion

During disasters such as bushfires, people are required to make decisions to protect their safety under time-sensitive, uncertain and stressful conditions. To support this, they mobilise a range of resources, including social resources, to assist with coping and decision making. Low levels of access to resources are widely understood to increase vulnerability through reduced coping, a component of models such as the Pressure and Release (PAR) model, in which risk of poor outcomes is considered a function of the intersection of hazard exposure and vulnerability [111]. The beneficial role of social resources in coping with disasters is widely documented in the disaster and stress literature [41,48,57], and having access to social resources is recognised as central to coping with disasters in theories of both disaster resilience and disaster vulnerability (e.g., [112]).

The current study aimed to explore the effects of different types of social resources on distress and coping self-efficacy during the 2019–2020 Black Summer bushfires in Australia, measured retrospectively in the months after the disaster, and to examine a wider range of types of social resources than has been examined in previous studies. The findings align with the theoretical and empirical literature supporting the role of social resources, particularly perceived social support, in the stress and coping process [49,50] and supporting better psychological outcomes after disasters [47,73]. In addition, they provide insight into how different types of social resources may protect against distress and promote psychological resources such as coping self-efficacy, which can assist in responding during a disaster. A key finding from this study is that the positive association between access to social resources and key psychological outcomes during bushfires—coping self-efficacy and distress levels—was observed for some types of social resources but not others.

### 4.1. Coping Self-Efficacy

Consistent with previous findings, gender was the strongest predictor in the coping self-efficacy model, with men being more likely to feel confident they could make decisions and cope with the bushfires compared to women [103]. However, age did not significantly predict coping self-efficacy, which is inconsistent with previous research suggesting that older people may have lower distress due to growth in coping self-efficacy because of previous experiences of natural hazards [44,61,104,113]. Higher perceived bushfire impact was also associated with lower coping self-efficacy.

Having access to three of the five types of social resources examined was associated with higher levels of coping self-efficacy during the bushfires: emotional support, practical support and lower loneliness. However, having higher levels of bushfire reciprocal support, or a higher sense of belonging, was not associated with higher coping self-efficacy. This indicates that having higher perceived access to social support and the psychological or tangible resources that these social networks provide was important for feeling confident in responding to and coping with the potential impacts of the bushfires, while broader perceived social resources on a community level and reciprocal support exchanges did not significantly impact coping self-efficacy.

Sense of belonging and bushfire reciprocal support were not significant predictors of coping self-efficacy, but were predictors of distress, with belonging predicting lower distress, but reciprocal support predicting higher distress. The findings suggest that these larger-scale, more diffuse social resources, or receiving or providing support within the community, are not strongly related to personal levels of coping self-efficacy. Conversely, a person’s close bonding social networks—which are more directly relied upon for emotional and practical support—appear more important for enabling coping self-efficacy.

Overall, the findings build on the body of evidence around the positive role of social resources in self-efficacy in disaster recovery [29], particularly perceived social support and bonding social capital. They also suggest a need to carefully define which types of social resources are in operation in a community, as not all types examined are associated with higher coping self-efficacy. The findings align with the COR model [20] and SCT [25,26], being consistent with the hypothesis that social resources are important for buffering against resource loss, reducing stress and increasing coping appraisal. However, the findings suggest that personal bonding social capital and cognitive social resources may be more strongly associated with positive psychological outcomes than other structural social resources. While the disaster practitioner literature often focuses on community-level social capital in disaster planning and risk reduction [52,114], our findings suggest there is a need to understand differences in access to bonding social capital and cognitive social resources such as perceived access to social support. Targeting investment in those with low access to bonding social capital is an important avenue for reducing the negative psychological and physical impacts of disasters and reducing underlying vulnerability to disasters.

### 4.2. Distress During Bushfires

In the second model, higher coping self-efficacy was the strongest predictor of lower distress. This is consistent with previous studies identifying a positive role for self-efficacy in reducing psychological distress after disasters and extends these findings to the disaster response stage [29,37,38,39]. This finding suggests that supporting resources that promote coping self-efficacy, specifically perceived access to social support and close bonding social connections (low levels of loneliness), identified as important in model 1, are likely to be important for reducing distress via increased coping self-efficacy. However, there are some inconsistent findings related to perceived access to social support, which we explore below.

As expected, the extent to which people were affected by the bushfires was a significant predictor of distress in both the regression models and bivariate analyses, with higher levels of self-rated bushfire impact predicting higher distress. This conforms with stress theory, suggesting that higher severity of bushfire impacts and resulting loss of resilience resources can reduce coping capacity and thereby increase distress [20,37,39].

Four of the five social resources had a significant association with distress, independent of coping self-efficacy. However, only two—sense of belonging and loneliness—had the expected relationship of higher social resources being associated with lower distress. These two social resources also differed from the other three in that respondents were asked about current resources at the time they completed the survey, rather than to recall levels during and immediately after the bushfires. The association between higher perceived social resources and lower distress and increased resilience is consistent with some past studies of disaster recovery [57,115], the broader stress and coping literature and COR theory [20,48].

Higher access to (i) emotional support and (ii) bushfire reciprocal support was associated with higher distress in model 2, while access to practical support was not a significant predictor of distress. These findings at first appear to contradict past work on the positive role of social support in coping with stress [20,48]. However, as model 2 examines the effects of social resources on distress after accounting for coping self-efficacy—which is known to have a positive effect on distress—we suggest the findings indicate that practical and emotional support and reciprocal support may primarily influence distress via their effect on coping self-efficacy, which could potentially mediate the relationship between access to these types of support and distress. There are several other possible explanations for the findings, which we discuss below; further investigation is needed to better understand which of these contribute to the findings, as there may be differing implications for practitioners, depending on the underlying factors resulting in the association.

Having access to emotional support and reciprocal support exchanges appears to have complex direct associations with distress, which may reflect previous findings and social support models in the literature. For example, past work has identified that those who experience more severe impacts from a stressor and experience heightened distress are more likely to receive support and, therefore, have higher perceived support [80,86]. That is, access to emotional and reciprocal support might not increase distress, but people who experience higher distress are more likely to access support or provide support to others. In this case, there is a need for practitioners to understand that access to the social resources is not causing the higher distress, but that the social resources in question need to be able to cope with provision of support to those experiencing heightened distress.

Another explanation, based on the Stress-Support Matching Hypothesis, is that those who experience greater impacts have higher expectations of receiving support and, thus, may perceive the support they receive to be inadequate—in other words, the resources provided are not sufficient to reduce stress [116]. The Social Support Effectiveness Model also suggests that the quantity, quality and effectiveness of support received can change its influence on distress [117]. For instance, receiving high levels of support that does not meet a person’s needs can increase their distress. For this reason, reciprocal support exchanges and perceptions of support may indirectly affect distress through the perceived quantity, quality and relevance of the support. If this explanation is correct, it suggests that practitioners should view heightened distress as a signal that the social support resources available to a person are not delivering appropriate support and that intervention may be needed to improve the quality of support provided.

It is also possible that the association between perceived emotional support and reciprocal support and distress relates to the reciprocal nature of support exchanges. According to the ‘social contagion’ effect, social networks can act to pass on distress (e.g., [46]). For example, Bryant et al. [46] found that after a bushfire, having contact with others who had experienced losses increased the likelihood of having PTSD. Therefore, accessing emotional and reciprocal support from people who have experienced losses and have high levels of distress may increase distress.

The findings of this study do not suggest that access to social support and reciprocal support exchanges are unimportant for coping among disaster-affected individuals, but rather that the relationships between social resources and psychological outcomes are complex and not always linear. Further work is needed to understand the influence of expectations of support; of the quality, quantity and relevance of social support; and who provides and receives support, as well as their effect on psychological responses during and immediately after disasters. The findings also point towards the potential importance of coping self-efficacy as a mediator of the relationship between social resources and distress. The potential complex relationships between social resources and psychological outcomes highlight the challenge of studying social resource dynamics and psychological experiences of disasters, including distress and coping-self-efficacy.

### 4.3. Limitations, Strengths and Future Directions

This study has some limitations. In particular, access to two types of social resources was measured around eight months after the bushfires ended, while the other three were measured based on recollections of levels of social resources that had been present during the bushfires. The two dependent variables were also retrospective measures asking for recollection of experiences during the bushfires eight months previously. In pilot testing, no participants indicated difficulty in recalling these experiences and states. It is important to note that there are ethical limitations to attempting to measure states during or immediately after a bushfire, with surveys of this type often inappropriate for delivery to communities immediately after they are impacted by bushfires. Single-item Likert scales were also used to measure access to emotional and practical support during the bushfires, which is a potential limitation, and there is a need for further research to develop measures of access to social support during disasters.

Both the coping self-efficacy and distress measures were based on a small number of items due to the length constraints of the survey. These measures require further development and validation to identify their utility for measuring bushfire-specific coping self-efficacy and distress. Additionally, the broad nature of the questions limits the ability to determine participants’ coping self-efficacy in relation to specific response actions such as evacuation. Moreover, people may feel they can cope with the impacts of disasters and feel confident in making decisions. However, perceived coping capacity may not necessarily result in the recommended behaviours. There was likely variability in people’s coping self-efficacy which could be attributed to factors not controlled for in this study such as education, as well as socioeconomic and personality factors. However, the broad nature of the questions allowed more participants to respond, contributing to the large sample size, a strength of this study.

As this study included a large sample of bushfire-affected people who experienced varying levels of bushfire impacts, the findings can be generalised to people affected by bushfires in Australia and may have relevance to those (i) affected by other natural hazards and (ii) in other countries that experience similar events.

Further research is needed to understand when and why different social resources have a positive association with the psychological resources that support responses during and after disasters. In particular, both quantitative and qualitative research are needed to gain a more in-depth understanding of how expectations of social support, the quality, quantity and types of reciprocal support exchanges, and perceived access to support and sense of connection differentially influence distress and coping self-efficacy during disasters. Moreover, further research is important to identify the potential for coping self-efficacy to mediate the effect of access to social resources on distress.

## 5. Conclusions

This study adds to the understanding of how social resources can support psychological responding during disasters through their role in increasing coping self-efficacy and lowering distress. Importantly, it suggests that some social resources are more strongly associated with high coping self-efficacy and lower distress, supporting the notion that different types of social resources are differentially related to psychological outcomes. Having higher perceived emotional and practical support was associated with increased coping self-efficacy, and lower levels of loneliness were associated with reduced distress and increased coping self-efficacy during bushfires. However, higher levels of perceived emotional support and bushfire reciprocal support were associated with higher distress after accounting for coping self-efficacy. Having a greater psychological sense of belonging was associated with lower distress but did not predict coping self-efficacy.

These findings reflect the distinct and complex role of social resources and reciprocal support exchanges. The finding that higher levels of emotional and reciprocal support was associated with higher distress does not indicate that they are not important but shows that the relationships between social resources and distress during disasters are complex and not always linear. For example, access to social resources or support exchanges during disasters may not always buffer against experiences of distress directly, but may do so through increasing coping self-efficacy to respond to and cope with the potential impacts of a disaster. The findings indicate that different types of social resources have the potential to increase positive psychological outcomes in response to disasters. However, they also highlight the potential limitations of their effectiveness, as it remains unclear under what conditions, and when these types of supports are most effective. The findings suggest that there is a need to consider other likely influences such as the perceived quantity, quality and relevance of support and the role of social networks in passing on distress, and that further research is needed in this area.

Our results suggest that supporting bonding social capital, including cognitive social resources such as sense of connection to others and reducing loneliness, may be important for disaster practitioners and individuals to enhance capacity to respond and cope during and immediately after disasters. The findings also suggest that those who provide support to others during a disaster may be at risk of experiencing higher levels of distress, and supporting coping self-efficacy is an important avenue for improving psychological response outcomes. The impact of different social resources on psychological responses during and after disasters, and the qualities that may moderate their effect, warrants further investigation.

## Figures and Tables

**Figure 1 ijerph-22-01341-f001:**
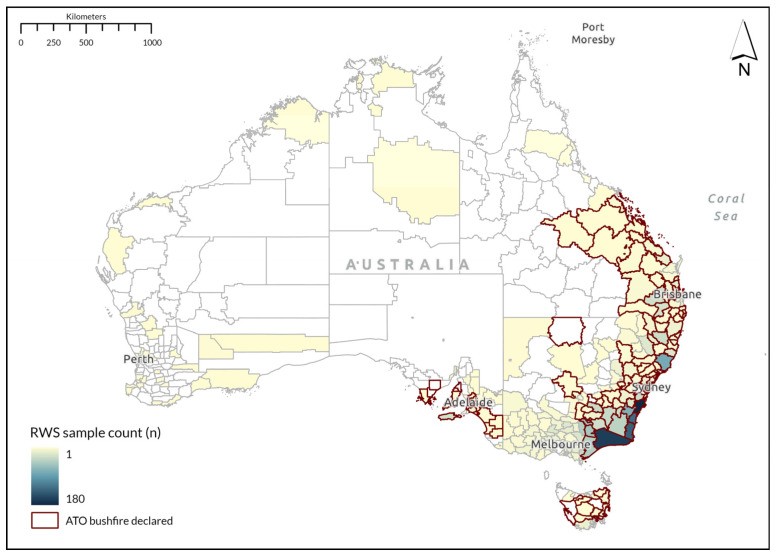
Map of sample by local government area (LGA).

**Table 1 ijerph-22-01341-t001:** Demographic characteristics of the sample.

Demographic Characteristic		n	% of Total Sample
Housing type	Own house outright	1424	54.5
	Live in a house with mortgage	787	30.1
	Renting	308	11.8
	Live in family’s home without paying rent	73	2.8
Household type	Sole person household	507	19.4
	Couple-only household	1153	44.2
	Couple parent with children household	605	23.2
	Single parent with children household	126	4.8
	Share or group household	94	3.6
	Other household type	126	4.8
Household financial prosperity	Very poor	26	1.0
	Poor	87	3.3
	Just getting along	606	23.2
	Reasonably comfortable	1229	47.1
	Very comfortable	591	22.6
	Prosperous	54	2.1

**Table 2 ijerph-22-01341-t002:** Individual survey questions and constructed variables used as dependent or independent variables for analysis.

Measure	Survey Items/Description of Scale	Response Scale
Demographic variables		
Age	How old are you?	Responses in individual years from ‘Under 18 years’ (excluded from analysis) to ‘100 or older’.
Gender	Do you identify as…	Female; male; other (e.g., non-binary, gender-fluid, no gender); prefer not to answer. Responses to ‘female’ and ‘male’ only were used in this paper due to low sample size for other responses.
Self-rated bushfire impact		
	Overall, how much were you and your household affected by any of the following in the last 12 months: Bushfire	7-point Likert scale from 1 (not at all affected) to 7 (very severely affected).
Dependent Variables		
Coping self-efficacy *	Three positively worded items formed the coping self-efficacy scale as the result of EFA (Table 3). The scale measures people’s retrospective confidence in making decisions about what to do, as well as how to protect themselves and their families and prepare for potential negative bushfire impacts, a form of self-efficacy in responding during the bushfires. High coping self-efficacy scores are theorised to indicate high levels of these psychological resources. Reliability analysis indicated that the scale had acceptable internal consistency (Cronbach’s Alpha = 0.76, see Table 3) [95].	The overall scale score was created by averaging the responses to three items, rated on a scale from 1 (strongly disagree) to 7 (strongly agree) in response to the following question: ‘How much do the following statements reflect how you felt during the period when bushfires were active’.
Distress during bushfire	Based on EFA, six negatively worded items formed a scale of negative psychological response experiences, which we named ‘distress during bushfire’. The six survey items were retrospective measures that asked about participant’s anxiety, sleeplessness and feelings of helplessness in protecting themselves and reducing their fire risk during the bushfires (see Table 3). This scale was theorised to measure psychological distress and perceived coping capacity related to bushfire response, and high scores reflect low levels of psychological resources that assist in responding. The scale showed high internal consistency according to Cronbach’s Alpha (Cronbach’s Alpha = 0.86, see Table 3).	The overall scale score was created by averaging the responses to the six items, measured from 1 (strongly disagree) to 7 (strongly agree) in response to the following question: ‘How much do the following statements reflect how you felt during the period when bushfires were active’.
Social resources		
Sense of belonging	The three-item sense of belonging scale used in the RWS was adapted from a scale published by Berry and Welsh [68]: I feel welcome here; I feel part of the community here; I feel like an outsider here (reversed). The scale was theorised to measure sense of belonging in community and demonstrates strong internal consistency (Cronbach’s Alpha = 0.86) [95,96]. Overall scale score was created by averaging the responses to items.	The overall score was calculated by averaging the responses on three items, measures on a 7-point scale 1 (strongly disagree) to 7 (strongly agree).
Bushfire reciprocal support	Based on EFA, the bushfire reciprocal support scale was formed using three RWS items specific to the experience of receiving and providing support during the bushfires (Table 3). The scales were calculated using the average of the individual survey items. Reliability analysis suggested that the scale had acceptable internal consistency (Cronbach’s Alpha = 0.74)	The overall scale score was created by averaging the responses to the three items, measured from 1 (strongly disagree) to 7 (strongly agree) in response to the following question: ‘How much do you agree or disagree with the following statements about how people in your LOCAL COMMUNITY have responded during and since the bushfires?’.
Emotional support	Single item: I had access to emotional support if I needed it, e.g., people I could talk to.	7-point scale 1 (strongly disagree) to 7 (strongly agree) in response to the following question: ‘How much do the following statements reflect how you felt during the period when bushfires were active’.
Practical support	Single item: I had access to practical support when I needed it, e.g., help getting my property prepared for fire.	7-point scale 1 (strongly disagree) to 7 (strongly agree) in response to the following question: ‘How much do the following statements reflect how you felt during the period when bushfires were active’.
Loneliness index	The loneliness measure used was a validated three-item scale, which has been used in multiple studies and has shown acceptable reliability [97,98]. The scale had high internal consistency (Cronbach’s Alpha = 0.89) and was created by averaging the responses to the following three loneliness items: ‘How often do you feel that you lack companionship?’; ‘How often do you feel left out?’; ‘How often do you feel isolated from others?’.	The overall score was calculated by averaging the responses on three items measured on a 5-point scale from 1 (never) to 5 (all of the time).

Note. * = Also a predictor variable in distress during bushfire model.

**Table 3 ijerph-22-01341-t003:** Exploratory factor analysis including results of constructed variables factor loadings, component eigenvalue, variance explained and reliability of scales.

Items on Scale	Factor Loadings		Component Eigenvalue		Variance Explained	Cronbach’s α
Coping self-efficacy (DV, IV)	1	2	3	2.20	18.28	0.76
I always or almost always felt confident to make decisions about what to do (R)		−0.62				
I felt confident I knew how to keep my loved ones safe (R)		−0.92				
I felt confident I could cope with the impacts of bushfire and smoke on my work or income (R)		−0.65				
Distress during bushfire (DV)				4.22	35.13	0.86
I often felt anxious or worried	0.80					
I sometimes found it hard to concentrate on anything	0.83					
I had periods of time where I slept poorly or had few hours of sleep	0.67		0.13			
I sometimes felt helpless to do anything to do anything to help people or places I care about	0.77		−0.10			
I sometimes felt there was nothing I could do to reduce the impacts of smoke on my household	0.67					
I sometimes felt there was nothing I could do to reduce the risk of fire causing damage to my home or house	0.57					
Bushfire reciprocal support (IV)				1.34	11.20	7.6
My neighbours and I helped each other out during the fires			0.79			
I was able to help others in my community during the fires			0.63			
I received help from others in my local community during the bushfires			0.71			

Note. (R) = Reverse-scored; All items were measured on a scale from 1 (strongly disagree) to 7 (strongly agree); DV = Dependent variables; IV = Independent variables.

**Table 4 ijerph-22-01341-t004:** Descriptive statistics for all variables measured on a 7-point or 5-point Likert scale (sample size, n = 2611).

Variables Measured on 7-Point Scale	Mean Score (1–7)	% Response		
		Low (Score of 1–3)	Moderate (Score of 4)	High (Score of 5–7)
Distress during bushfire (DV)	4.27	40.4	21.0	38.5
Coping self-efficacy (DV)	4.80	24.9	21.1	54.0
Practical support	4.22	34.7	17.6	47.7
Emotional support	4.75	23.8	14.6	61.6
Bushfire reciprocal support	4.45	30.9	24.7	44.4
Sense of belonging	5.65	9.7	13.7	76.8
Loneliness *	2.44	63.8	26.9	9.3
Self-rated bushfire impact	4.26	35.3	15.3	49.4

* Measured on a 5-point scale: low (score of 1–2), moderate (score of 3), high (score of 4–5).

**Table 5 ijerph-22-01341-t005:** Bivariate correlations between five social resources and (i) coping self-efficacy and (ii) distress during bushfire.

Variable	2. Distress During Bushfire	3. Practical Support	4. Emotional Support	5. Bushfire Reciprocal Support	6. Sense of Belonging	7. Loneliness	8. Age	8. Gender	9. Self-Rated Bushfire Impact
1. Coping self-efficacy	−0.36 ***	0.34 **	0.38 ***	0.09 ***	0.15 ***	−0.19 ***	0.10 ***	0.26 ***	−0.09 ***
2. Distress during bushfire		−0.05 *	−0.06 **	0.20 ***	−0.10 ***	0.28 ***	−0.15 ***	−0.26 ***	0.37 ***
3. Practical support			0.53 ***	0.24 ***	0.19 ***	−0.15 ***	0.01	0.03	0.01
4. Emotional support				0.20 ***	0.27 ***	−0.25 ***	0.08 ***	−0.01	0.01
5. Bushfire reciprocal support					0.23 ***	−0.10 ***	−0.02	−0.04 *	0.27 ***
6. Sense of belonging						−0.39 **	0.26 ***	0.01	0.03
7. Loneliness							−0.26 ***	−0.11 ***	0.04
7. Age								0.21 ***	0.04 *
8. Gender									−0.06 **

Spearman’s correlation coefficient, * *p* < 0.05, ** *p* < 0.01, *** *p* < 0.001.

**Table 6 ijerph-22-01341-t006:** Summary of hierarchical regression model predicting coping self-efficacy during the bushfires.

Variable	B[95% CI]	Beta	Sr^2^	t
Step 1				
Age	0.00 [−0.00, 0.01]	0.04	0.00	2.07
Gender	0.73 *** [0.62, 0.85]	0.24	0.06	12.51
Self-rated bushfire impact	−0.07 *** [−0.10, −0.04]	−0.09	0.01	−4.69
Step 2				
Age	0.00 [−0.00, 0.01]	0.01	0.00	0.46
Gender	0.74 *** [0.62, 0.83]	0.24	0.05	13.52
Self-rated bushfire impact	−0.07 *** [−0.10, −0.04]	−0.09	0.01	−4.92
Practical support	0.15 *** [0.11, 0.18]	0.19	0.02	9.08
Emotional support	0.20 *** [0.17, 0.24]	0.25	0.04	12.05
Bushfire reciprocal support	0.01 [−0.03, 0.04]	0.01	0.00	0.28
Sense of belonging	0.01 [−0.03, 0.06]	0.01	0.00	0.61
Loneliness	−0.09 ** [−0.15, −0.03]	−0.06	0.00	−3.03

Note: Listwise n = 2611; CI = confidence interval; Sr^2^ = squared semi-partial correlations; “Gender” (dummy coding: F = 0, M = 1). *** *p* < 0.001; ** *p* < 0.01; * *p* < 0.05.

**Table 7 ijerph-22-01341-t007:** Summary of hierarchical regression model predicting distress during the bushfires.

Variable	B[95% CI]	Beta	Sr^2^	t
Step 1				
Age	−0.01 *** [−0.02, −0.01]	−0.12	0.01	−6.52
Gender	−0.73 *** [−0.84, −0.61]	−0.22	0.04	−12.10
Self-rated bushfire impact	0.32 *** [0.29, 0.35]	0.36	0.13	20.42
Step 2				
Age	−0.01 * [−0.01, −0.00]	−0.04	0.00	−2.32
Gender	−0.43 *** [−0.55, −0.32]	−0.13	0.01	−7.64
Self-rated bushfire impact	0.25 *** [0.22, 0.28]	0.28	0.07	16.70
Practical support	0.02 [−0.02, 0.05]	0.02	0.00	1.06
Emotional support	0.07 *** [0.03, 0.10]	0.08	0.00	3.81
Bushfire reciprocal support	0.15 *** [0.12, 0.19]	0.16	0.02	9.09
Sense of belonging	−0.07 ** [−0.11, −0.02]	−0.05	0.00	−2.76
Loneliness	0.35 *** [0.29, 0.41]	0.21	0.03	11.69
Coping self-efficacy	−0.32 *** [−0.36, −0.28]	−0.31	0.07	−16.12

Note: Listwise n = 2611; CI = confidence interval; Sr^2^ = squared semi-partial correlations; “Gender” (dummy coding: F = 0, M = 1). *** *p* < 0.001; ** *p* < 0.01; * *p* < 0.05.

## Data Availability

The raw data supporting the conclusions of this article will be made available on request due to privacy and ethical restrictions.

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
