# Peer review of "The Impact of Different Types of Social Resources on Coping Self-Efficacy and Distress During Australia’s Black Summer Bushfires"

_ijerph, 2025, doi:10.3390/ijerph22091341_

Round 1

Reviewer 1 Report

Comments and Suggestions for Authors

Dear authors. Kindly note the followings:

  1. The paper does not provide a sound contribution to the current knowledge as social resources and psychological impacts of natural disasters have been discussed in the literature extensively.
  2. There is a mix between several concepts without clearly discussing the differences. What you refer to as networking during disasters relates mainly to crisis communications, which was not adequately discussed in the paper. 
  3. In page 4 of 23, the paragraph starting from line 160-165 should be removed. It contradicts all the above discussion and suddenly jumps into a negative discussion disregarding all the positive benefits that were previously discussed. 
  4. The results presented are very basic and simple and can be easily anticipated. Kindly note that this type of research requires more detail and insight from behavioral sciences, which was not adequately presented in the paper.
  5.  There is inadequate discussion on the practical implications that can reflect on best practices in the field, making the social and practical implications theoretically valid only without emphasizing the real-world lessons that can be learned.

Author Response

Please see document attached for responses to reviewer 1's comments.

Reviewer 2 Report

Comments and Suggestions for Authors

I recommend acceptance with minor revisions. The authors present a clear experimental design with an interesting discussion that provides academic relevance, observing that the relationships are complex and not always linear with different social resources associated with different psychological outcomes and specific influences of social networks. Please find specific comments below.  

Abstract

L16. If you mention the two psychological outcomes, it is necessary to mention the five social resources. 

Introduction

L20. Please mention your most relevant statistical results for the variables.

L40. You need some references to provide support for the affirmations presented. 

L83-94. A table with the resilience resources could be better for presenting the list. 

L241. Please elaborate on the characteristics of the waves, as it is mentioned that this is Wave 8. 

L305. A map with the distributions would help for a better visualization of the data. 

L314. First, present the data available from Table 1 and the after data from Table 2. 

Table 1. Usually, demographic variables are presented before the surveys. 

Author Response

Please see attached document for responses to reviewer 2's comments.

Reviewer 3 Report

Comments and Suggestions for Authors

It is worthwhile to study social resources that can predict the coping efficacy of wildfire victims. However, it seems necessary to further supplement the manuscript in a way that can secure the internal validity of the study scientifically. Here are somethings I would like you to improve it:

  1. In the introduction, you need to present a more specific rationale for how the predictors you assumed can be related to the criterion variable.

  1. Please clarify how far previous studies have been conducted on this topic and how this study can fill the research gap.

  1. I understand that it was inevitable to develop items because there is no validated scale yet. Even so, if it is not a scale development study, there is no need to present the EFA results in a table. Instead, it seems that it would be sufficient to explain the results in a sentence explaining the scale and provide examples of the items. And for the explanation of the tools that measured each variable, you can explain the reliability or validity of each scale instead of a table.

  1. Since the values measured by single Likert items are not parameters, in principle, parametric statistical analysis such as correlation analysis or regression analysis cannot be performed. If this is unavoidable, it should be disclosed in the explanation of statistical analysis and also described in the limitations of the study.

  1. Does this mean that you created dummy variables by using a 5-point scale of “low (score of 1-2), moderate (score of 3), and high (score of 4-5)”? It is inappropriate to input data dummy variables in this way into regression analysis.

  1. Gender and age were adjusted as covariates, but a rationale must be provided as to whether they meet the conditions for exogenous confounding variables that must be adjusted as covariates. The conditions must be related to the criterion variable (dependent variable) as well as the predictor variable (independent variable). If they do not meet the conditions and are included as covariates, the model's simplicity will decrease and the regression model will be distorted due to Berkson's paradox.

Author Response

Please see attached document for responses to reviewer 3's comments.

Reviewer 4 Report

Comments and Suggestions for Authors

Title:
The current title does not accurately reflect the study’s comparative approach. Since the study investigates the distinct roles of five types of social resources, a more suitable title would be:
“The Impact of Different Types of Social Resources on Coping Self-Efficacy and Psychological Distress During the Black Summer Bushfires.”

Abstract:

  • Consider adding a brief definition or illustrative examples of “social resources” at the beginning (e.g., emotional support, practical support, sense of belonging) to ground the reader in the study's key construct.
  • The sentence “little is known about the unique influence of different social resources on distress and coping” implies a comparative aim, which should be better reflected in the title and objective statements.
  • Please explicitly list the five types of social resources examined in the abstract.
  • Report key statistical results (e.g., p-values, correlation coefficients or beta values) in the abstract to enhance objectivity.
  • The conclusion needs to be more definitive and summarize key findings clearly.

Introduction:

  • Clarify the opening statement regarding natural hazards. Cite references specific to bushfires and climate change-induced wildfire frequency (e.g., ref [2] is climate-specific).
  • Include a reference to support the statement: “supporting at-risk populations to respond effectively... can reduce severity...”
  • Reconsider use of the phrase “during and immediately after,” especially since the data were collected months later. The term “post-disaster” would be more accurate.
  • A conceptual framework diagram illustrating how theories such as COR and SCT relate to social resources, coping self-efficacy, and distress would improve comprehension.
  • The claim that “social resources are widely documented...” requires a citation with systematic reviews or empirical studies supporting that assertion.
  • References 39–42 should be limited to original studies that demonstrate the impact of social resources on psychological outcomes.
  • Section 1.1 should be subdivided into smaller subheadings categorizing social resources (e.g., emotional, practical, cognitive, structural) to improve readability.

Methodology:

  • Sections 2.1 and the opening paragraph of the methodology are largely descriptive and do not contribute to the methodological rigor—consider shortening or removing them.
  • The sampling design is weak. Recruitment through social media and general mailing lists lacks verification of participants’ exposure to the bushfire event. There is no systematic sampling frame, and self-identification as “affected” is not verifiable.
  • Clearly specify the regions oversampled (e.g., NSW, VIC) rather than stating “rural and remote regions.”
  • The term “during the bushfire” is misleading since data collection was months later. “Post-disaster recall” or “retrospective self-reporting” is more accurate.
  • There should be a clearer outline of:
    • Sample size calculation (if any),
    • Inclusion and exclusion criteria (none mentioned),
    • Verification of exposure status.
  • Measurement tools are problematic. Creating new items and using EFA without validation limits comparability and reliability. Standardized tools for distress (e.g., DASS, GAD-7) and coping (e.g., Brief COPE, CSE scale) should have been used. Novel measures should undergo validation before full-scale deployment.

Results & Discussion:

  • Participant demographics should include socioeconomic status, housing type, family structure, etc., to understand the context of their coping capacity.
  • Since Likert scales were used, descriptive results should include mean and SD rather than arbitrary low/moderate/high categories.
  • The correlation coefficients are mostly <0.5, indicating weak to moderate relationships—this should be acknowledged.
  • The discussion section lacks depth in explaining why certain social resources increased or failed to decrease distress. The authors should critically interpret why emotional and reciprocal support correlated positively with distress, especially after controlling for coping self-efficacy

Author Response

Please see attached document for responses to reviewer 4's comments.

Reviewer 5 Report

Comments and Suggestions for Authors

This article highlights an important aspect of disaster resilience. It’s a valuable contribution because it shows how social resources not only buffer distress but also empower people to believe in their own coping abilities.

The authors have reviewed the relevant literature comprehensively and the manuscript provides a strong justification for the study’s aims.

The description of the study methods is clear and detailed, with all variables appropriately described.

The authors have presented the results clearly, using suitable statistical analyses. The tables are well organized, and the models are described in sufficient detail.

The authors have provided a clear and balanced discussion that relates their results to previous research and theoretical implications. The Conclusion provides a strong, relevant closing to the manuscript.

The manuscript follows the proper referencing format as required by the journal.

In my opinion, the manuscript is of publishable quality and does not require major revisions.

Author Response

Please see attached document for responses to reviewer 5's comments.

Round 2

Reviewer 1 Report

Comments and Suggestions for Authors

Dear authors;

I see that you have made some substantial corrections and fine-tunings to the previous version and therefore the paper has improved. I only have two comments. As long as you are discussing psychological impacts and resource mobilization, it becomes recommended to incorporate Wisner's view (pressure and release model) in your discussion (just briefly) to strengthen the theoretical foundation of the research. Please refer to (@Risk) for more detailed information. I have also spotted some weakly worded statements, e.g. "Having access to resources that support wellbeing is important for supporting functioning in responses to stressors such as disasters", please rephrase. 

Author Response

Comment 1: I see that you have made some substantial corrections and fine-tunings to the previous version and therefore the paper has improved. I only have two comments. As long as you are discussing psychological impacts and resource mobilization, it becomes recommended to incorporate Wisner's view (pressure and release model) in your discussion (just briefly) to strengthen the theoretical foundation of the research. Please refer to (@Risk) for more detailed information. I have also spotted some weakly worded statements, e.g. "Having access to resources that support wellbeing is important for supporting functioning in responses to stressors such as disasters", please rephrase. 

Response 1: We thank the reviewer for noting that the paper has improved after the revisions were made. We have now added a brief discussion of the Pressure and Release Model at the start of the discussion, this is discussed in relation to the importance of resources for reducing vulnerability to disasters (see rows 522 to 528). We have also edited wording and sentence structure throughout the manuscript, including the statement the reviewer identified as weakly worded. 

Reviewer 3 Report

Comments and Suggestions for Authors

Thank you for revising the manuscript in response to my comments in the first review. You also provided some detailed explanations, which I found very convincing as a reviewer.

If you continue to carefully proofread and improve the quality of the manuscript, I believe it will be acceptable for publication in an academic journal.

Author Response

Comment 1: Thank you for revising the manuscript in response to my comments in the first review. You also provided some detailed explanations, which I found very convincing as a reviewer.

If you continue to carefully proofread and improve the quality of the manuscript, I believe it will be acceptable for publication in an academic journal.

Response 1: Thank you for your positive words and for your suggestions. We have now carefully edited the wording and sentence structure and fixed minor errors throughout the manuscript to improve the overall quality of the manuscript.

Reviewer 4 Report

Comments and Suggestions for Authors

authors have addressed the comments and made necessary amendments

Author Response

Comment 1: authors have addressed the comments and made necessary amendments

Response1: Thank you for your comment. We have now carefully edited the manuscript to fix minor errors and improve phrasing and sentence structure based on other reviewers' suggestions.